# Novel D-A-π-A1 Type Organic Sensitizers from 4,7-Dibromobenzo[*d*][1,2,3]thiadiazole and Indoline Donors for Dye-Sensitized Solar Cells

**DOI:** 10.3390/molecules27134197

**Published:** 2022-06-29

**Authors:** Nikita S. Gudim, Ekaterina A. Knyazeva, Ludmila V. Mikhalchenko, Maksim S. Mikhailov, Lu Zhang, Neil Robertson, Oleg A. Rakitin

**Affiliations:** 1N. D. Zelinsky Institute of Organic Chemistry, Russian Academy of Sciences, 119991 Moscow, Russia; nikitosgudim@gmail.com (N.S.G.); katerina_knyazev@mail.ru (E.A.K.); mlv@ioc.ac.ru (L.V.M.); 2Nanotechnology Education and Research Center, South Ural State University, 454080 Chelyabinsk, Russia; mihailov_maxim_s@mail.ru; 3EaStCHEM School of Chemistry, University of Edinburgh, Edinburgh EH9 3FJ, UK; luzhang2019@163.com

**Keywords:** benzo[*d*][1,2,3]thiadiazole, Suzuki cross-coupling reaction, D-A-π-A1 dyes, dye-sensitized solar cells, optical properties, electrochemical properties, power conversion efficiency

## Abstract

Two novel D-A-π-A1 metal-free organic dyes of the **KEA** series containing benzo[*d*][1,2,3]thiadiazole (isoBT) internal acceptor, indoline donors fused with cyclopentane or cyclohexane rings (D), a thiophene as a π-spacer, and a cyanoacrylate as an anchor part were synthesized. Monoarylation of 4,7-dibromobenzo[*d*][1,2,3]thiadiazole by Suzuki-Miyamura cross-coupling reaction showed that in the case of indoline and carbazole donors, the reaction was non-selective, i.e., two monosubstituted derivatives were isolated in each case, whereas only one mono-isomer was formed with phenyl- and 2-thienylboronic acids. This was explained by the fact that heterocyclic indoline and carbazole fragments are much stronger donor groups compared to thiophene and benzene, as confirmed by cyclic voltammetry measurements and calculation of HOMO energies of indoline, carbazole, thiophene and benzene molecules. The structure of monoaryl(hetaryl) derivatives was strictly proven by NMR spectroscopy and X-ray diffraction. The optical and photovoltaic properties observed for the **KEA** dyes showed that these compounds are promising for the creation of solar cells. A comparison with symmetrical benzo[*c*][1,2,3]thiadiazole dyes **WS-2** and **MAX114** showed that the asymmetric nature of benzo[*d*][1,2,3]thiadiazole **KEA** dyes leads to a hypsochromic shift of the ICT band in comparison with the corresponding benzo[*c*][1,2,5]thiadiazole isomers. **KEA** dyes have a narrow HOMO-LUMO gap of 1.5–1.6 eV. Amongst these dyes, **KEA321** recorded the best power efficiency (PCE), i.e., 5.17%, which is superior to the corresponding symmetrical benzo[*c*][1,2,3]thiadiazole dyes **WS-2** and **MAX114** (5.07 and 4.90%).

## 1. Introduction

Recently, the problem of the increasing temperature of the earth has become seriously aggravated, largely due to the use of carbonized energy sources such as coal, oil and gas. The second cause of concern for humanity is the depletion of these energy sources. Therefore, the search for renewable and environmentally friendly energy sources has become an important and urgent task. Solar energy is the most accessible and practically inexhaustible source of energy. In the past few decades, along with well-developed silicon-based photovoltaic devices, research on dye-sensitized solar cells (DSSCs) has been intensively developed due to their relative cheapness, low dye consumption, low-light capability and facile fabrication [1,2,3,4]. The photosensitizer is the most important component of this type of solar cell, as it directly determines their efficiency. There are three main classes of sensitizers that have shown the most outstanding results: ruthenium complexes [5,6,7], porphyrin dyes [8,9,10] and metal-free organic dyes [11,12,13]. Among them, metal-free organic dyes seem to be the most promising due to their tunable optoelectronic properties via flexible molecular designs, high molar extinction coefficients for light harvesting, low synthetic and purification cost compared to their counterpart, i.e., ruthenium and porphyrin dyes, high molar extinction coefficient, easy synthesis, low purification cost and virtually unlimited structure variation, as required to achieve the best possible performance [11,12,13,14,15]. Most of these molecules have the traditional D-π-A structure, consisting of an electron donor (D), a π-bridge and an electron acceptor (A) [16,17,18].

About ten years ago, it was found that structures like D-A-π-A1 possess higher light-harvesting ability and better power conversion efficiency (PCE) and stability [19,20,21]. One of the most common auxiliary acceptors used in D-A-π-A1 structures was benzo[*c*][1,2,5]thiadiazole (BT) [22,23,24,25]. Surprisingly, its isomer, benzo[*d*][1,2,3]thiadiazole (isoBT), has not been explored in DSSCs. It was previously found that the most suitable precursor, 4,7-dibromobenzo[*d*][1,2,3]thiadiazole, can be easily prepared in a two-step synthesis process from the commercially available 2-aminobenzothiol [26,27]. A study of its chemical properties showed that the nucleophilic substitution of 4,7-dibromobenzo[*d*][1,2,3]thiadiazole with morpholine gave selectively mono-4-substituted 4-(7-bromobenzo[*d*][1,2,3]thiadiazol-4-yl)morpholine [28]. In addition to the cross-coupling reactions of this dibromide, 4,7-di(hetaryl) derivatives can be easily prepared in moderate yields by Stille reaction [27], and some 4,7-di(hetaryl)benzo[*d*][1,2,3]thiadiazoles can lead to compounds with promising photovoltaic properties. Therefore, in this work, we aimed to obtain new organic dyes based on the benzo[*d*][1,2,3]thiadiazole (isoBT) unit. Herein, we describe a study of cross-coupling reactions of 4,7-dibromobenzo[*d*][1,2,3]thiadiazole with donor molecules in order to obtain monosubstituted derivatives for the preparation of donor-acceptor-π-spacer-acceptor1 (D-A-π-A1)-conjugated molecules which could find potential applications in DSSCs. In these molecules, benzo[*d*][1,2,3]thiadiazole played the role of internal acceptor, a thiophene cycle as a π-bridge and a cyanoacrylate residue as a terminal acceptor and anchor.

## 2. Results and Discussion

### 2.1. Synthesis of D-A-π-A1 Dyes

The synthetic strategy for the preparation of the target D-A-π-A1 dyes consisted of a number of steps, as shown in Figure 1. It included two successive cross-coupling reactions with boronic acids or their esters (Suzuki-Miyamura reaction) and with tributylstannyl derivatives (Stille reaction), followed by the Knoevenagel reaction with cyanoacetic acid *tert*-butyl ester, and finally, hydrolysis of ester **4** to afford the desired **KEA** dyes.

Monoarylation of 4,7-dibromobenzo[*d*][1,2,3]thiadiazole is the most challenging task and requires special attention in order to achieve the best yields [25]. According to the literature [[25] and references therein], the two most common protocols for the substitution of bromine atoms with aryl groups are the reaction with arylboronic acids or its esters (the Suzuki reaction) and the reaction with trialkylstannates (the Stille reaction). Arylation of 4,7-dibromobenzo[*c*][1,2,5]thiadiazole with one equivalent of trialkylstannyl derivative led to low yields (15–35%) of mono-arylated heterocycles, since they reacted readily further with the formation of bis-aryl products [24]. Better results for the isolation of mono-substituted derivatives have been achieved by Suzuki cross-coupling reaction; for example, monoarylated derivatives have been isolated in high yields in the reactions of 4,7-dibromobenzo[*c*][1,2,5]thiadiazole with arylboronic acid by using the same NCP pincer palladacycle [29]. Also where the reactivity of two bromine atoms in 4,7-dibromo[1,2,5]chalcogenadiazolo[3,4-*c*]pyridines has differed substantially, regioselective arylation occurred [30,31] and the Suzuki reaction gave a considerably higher yield (72%) compared to the Stille reaction (27%) [24].

Therefore, the behavior of 4,7-dibromobenzo[*d*][1,2,3]thiadiazole **1** in the palladium-catalyzed Suzuki-Miyamura coupling reactions in order to obtain mono-aryl(hetaryl)benzo[*d*][1,2,3]thiadiazoles was investigated (Figure 2). This study, with one equivalent of (9-hexyl-9*H*-carbazol-3-yl)boronic acid and its pinacolate ester **2**, included varying the base, solvent and reaction temperature. The results are summarized in Table 1.

A study of the Suzuki reaction of 4,7-dibromobenzo[*d*][1,2,3]thiadiazole **1** with one equivalent of boronic acid **2a** or its pinacolate ester **3a** showed that the reaction was not selective and led to a mixture of two monohetaryl derivatives **7a** and **8a** and bis-aryl derivative **9a**, with yields of these products of around 20%. The nature of the solvent, either type of solvent (THF) or a mixture of solvents that are capable of solubilizing organic substrates (THF, toluene, or dioxane) and inorganic salts (water) did not influence the results of the reaction. The yields for the pinacolate ester **3a** were slightly higher than for the corresponding boronic acid **2a** (compare Entries 1 and 8 in Table 1). THF gave the best yields of mono-substituted derivatives **7a** and **8a** (Table 1, Entry 1).

The conditions for the synthesis of monohetaryl derivatives **7a** and **8a** were further used for several boronic acids **2** and esters **3** (Figure 3). It was found that in the case of indoline donors **2b-d**, the reaction remained non-selective; two monosubstituted derivatives, **7b-d** and **8b-d**, were isolated in all these cases with yields of about 20% (Table 2). Compounds **8c**, **8d** and **9d** were not isolated in a pure state due to the difficulty of purification; nevertheless, their structure was proven by ^1^H NMR and MALDI-MS spectral data. Unexpectedly, when phenyl- and 2-thienylboronic acids **6e,f** (one equiv) were involved in Suzuki-Miyamura coupling reactions with 4,7-dibromobenzo[*d*][1,2,3]thiadiazole **1**, only one mono-isomer, **7e,f**, was formed together with disubstituted derivatives **9e,f** with double yields (compared to the yields of mono-products **7a-d**), i.e., about 40% (Figure 3).

As is known [32,33], the Suzuki reaction mechanism includes the formation of an intermediate complex of palladium with halide and boronic acid, in which electrons are transferred from the metal to the halide and from the boronic acids to the metal. In order to evaluate the donor ability of the substituent of the studied boronic acids to donate electrons, the oxidation potentials of donor **10** are shown in Table 3, as determined by cyclic voltammetry (CV) (see Appendix A). It was established that all compounds were oxidized irreversibly on platinum and glassy carbon electrodes, and the values of the oxidation potentials practically did not depend on the type of electrode. Oxidation peaks of donors **10a,c,d** were observed by CV in DMF solutions (0.1 M Bu_4_NClO_4_), whereas for **10e** and **10f,** having a higher oxidation potential, peaks were obtained in acetonitrile solution. To estimate the energies of the highest occupied molecular orbital (E_HOMO_), we used estimated peak onset values (E^ox^_onset_). The values of E^ox^_onset_ were calculated relative to the potential of the reversible oxidation of ferrocene/ferrocenium (Fc/Fc^+^) redox pair, the absolute potential of which was taken as −5.1 eV [34,35]. The values of E_HOMO_ presented in the Table 3 were calculated according to Equation (1) (see below).

The cyclic voltammetry showed that heterocyclic fragments **10a**, **10c** and **10d** had much higher E_HOMO_ values, which indicated that they were much stronger donor groups compared to benzene **10e** and thiophene **10f**. It can be assumed that the relative ease of electron transfer in the intermediate complex may be one of the factors contributing to the formation of a mixture of monoisomers **7a-d** and **8a-d**, in contrast to thienyl- and phenylboronic acids **2e,f**.

### 2.2. Proof of the Structure of Monobromo Derivatives **7** and **8** by NMR Spectroscopy and X-ray Diffraction

A set of two-dimensional NMR spectra was recorded in order to correlate the signals in the ^1^H and ^13^C spectra. The key interatomic interactions for both isomers **7b** and **8b** are shown in Figure 1. For product **8b**, a proton doublet at C6 is observed at 7.53 ppm with an interaction in HMBC spectra with the quaternary carbon 8 of the donor fragment (highlighted in red), while for the second proton of the isothiadiazole ring at C5 at 7.83 ppm, this interaction is absent. An additional confirmation of this proved the interaction in HMBC spectra of protons at C9 and C11 of the donor fragment with the quaternary carbon 7 of isothiadiazole (highlighted in green). Similarly, for isomer **7b**, key interactions in HMBC spectra of the proton at C5, which manifests itself at 7.55 ppm, with the quaternary carbon 8 of the donor (red highlight), as well as protons at C9 and C11 of the donor fragment with carbon 4 of isothiadiazole (green highlight), are observed. The absence of cross peaks in the HMBC spectrum of compound **7b** related to the interaction of the proton at C6 and the nodal carbon 8 of the donor, as well as the absence of NOE interaction between protons at C6 and C9/C11 as well as the presence of one between the protons C5 and C9/C11 (blue arrows) additionally confirm the correlation of signals.

The use of NMR spectroscopy made it possible to combine the results of XRD and HMBC data, and perform a complete correlation of signals in the ^1^H and ^13^C spectra for dibromide **1** (Figure 1) and isomers **7b**, **8b** (see Appendix A). As such, we were able to establish that the introduction of a donor substituent led to an upfield shift of the signals from both the nearest proton and the corresponding carbon of the benzo[*d*][1,2,3]thiadiazole system compared to the parent 4,7-dibromobenzo[*d*][1,2,3]thiadiazole **1**. In this case, the signal of carbon, which had undergone substitution, was shifted to a lower field, and the signal of carbon with the remaining bromine atom had a chemical shift ±3 ppm relative to the corresponding dibromide **1** chemical shift (Table 4). At the same time, the chemical shifts of carbons bearing a bromine atom (C7 for **7**; C4 for **8**) were characterized by the largest upfield shifts compared to the rest of the signals of aromatic hydrocarbons. Based on the above conclusions, an unambiguous determination of the structure of all obtained mono-substituted compounds **7** and **8** was made.

For compounds **8c**, **8d**, **9d**, which could not be isolated in pure form, ^1^H NMR spectra, as well as MALDI-TOF spectra were recorded (see Appendix A), on the basis of which their formation could be confirmed. Thus, for compound **8d**, the doublet corresponding to the proton at C6 appeared at 7.65 ppm, and the proton doublet at C5 was observed at 7.85 ppm, while the proton signals at C5/6, recorded for isomer **7d**, were observed as a minor product. Additional confirmation could serve as a change of chemical shifts of protons at the nodal carbon atoms 12/12′ of the donor fragment: for **7d**, two multiplets were observed in the regions of 3.92–4.01 and 4.86–4.93 ppm, and for **8d** for the main product, in the regions of 4.65–4.74 ppm and 5.37–5.45 ppm. In this case, the ^1^H NMR spectrum of the bis-derivative **9d** was characterized by the presence of all four multiplets in the given regions, integrating as single protons.

In the case of compound **8c**, the difference in chemical shifts of key protons at C5/6 was not so characteristic, since the chemical the shift of one of the doublets was the same for both isomers; however, the doublet signal of the second proton for **8c** appeared in a weaker field, at 7.86 ppm corresponding to a proton at C6, and a doublet at 7.86 ppm characterized the proton at C5. In the high-field region of proton signals at C12/12’, the difference in chemical shifts for both isomers was not as critical as in the case of the **7d–8d** pair, but its presence also served as additional evidence for the formation of the **8c** isomer.

The structures of a few monobromo derivatives **7** and **8** were confirmed by single crystal X-ray diffraction study of compounds **7b**, **8b**, **7e** and **7f** (Figure 2). 

### 2.3. Synthesis and Characterization of the Dyes

Synthesis of the target dyes in the **KEA321** and **KEA337** series was carried out using Stille cross-coupling reaction of 4-substituted 7-dibromobenzo[*d*][1,2,3]thiadiazole with π-spacer **5** (Figure 4). This stage was carried out under conditions of Stille reactions catalyzed by PdCl_2_(PPh_3_)_2_ in toluene followed by saponification with HCl to remove dioxolane protection. After this, the products were introduced into the Knoevenagel reaction with cyanoacetic acid *tert*-butyl ester with subsequent hydrolysis resulting in the final structures **KEA321** and **KEA337**, which, after thorough chromatographic purification, were used in solar cell device fabrication.

### 2.4. Optical Properties

For the obtained dyes **KEA321** and **KEA327**, UV-Vis spectra were recorded in a DCM solution at a concentration of 5.5 × 10^−5^ M. Table 5 shows the absorption maxima (λ_max_) and the corresponding extinction coefficients (ε) for **KEA** dyes, as well as for isomeric dyes **MAX114** and **WS-2** based on benzo[*c*][1,2,5]thiadiazole (Figure 3).

Both new dyes had two absorption maxima; however, for the **KEA337** compound, due to the proximity of both maxima, their merging was observed. The short-wavelength absorption band at 400 nm, which for **KEA337** was identified as a shoulder, and for **KEA321** was characterized by a high extinction coefficient, corresponded to π/π* electron transition, and long-wavelength absorption maxima in the range of 450–500 nm indicated the presence of an intramolecular charge transfer (ICT) process between donor and acceptor fragments (Figure 4). For all four dyes, π/π* transition absorption peaks were observed in a narrow wavelength range, i.e., 394–401 nm, which characterizes close values of the energy level gap. The long-wavelength absorption maxima obeyed the previously established regularity [27]: the asymmetric nature of benzo[*d*][1,2,3]thiadiazole leads to a hypsochromic shift of the ICT band in comparison with the corresponding benzo[*c*][1,2,5]thiadiazole isomers. Higher extinction coefficients of dyes based on benzo[*c*][1,2,5]thiadiazole may indicate that the symmetry breaking of the acceptor benzo[*d*][1,2,3]thiadiazole fragment reduces the efficiency of intramolecular charge transfer for compounds of the **KEA** series.

### 2.5. Electrochemical Properties

To estimate the energy values of the frontier orbitals and to determine the stability of the particles formed during electron transfer, cyclic voltammetry patterns (CV curves) of **KEA321** and **KEA337** were measured. The first stages of the electrooxidation (EO) and electroreduction (ER) curves of the studied compounds are shown in Figure 5, and the potential values and calculated frontier orbitals energies are summarized in Table 6. 

The potentials of the ER and EO peaks of **KEA321** and **KEA337** were close to each other, the ER in both cases proceeded irreversibly, and the EO peak of **KEA321** had a quasi-reversible character, in contrast to the irreversible EO peak of **KEA337** (Table 6). To calculate the energies of the lowest unoccupied molecular orbital (E_LUMO_) and the highest occupied molecular orbital (E_HOMO_), we used estimates of the peak onset values ER (E^red^_onset_) and EO (E^ox^_onset_), respectively. The values of E^red^_onset_ and E^ox^_onset_ were calculated relative to the potential of the reversible oxidation of ferrocene/ferrocenium (Fc/Fc^+^) redox pair, the absolute potential of which equaled −5.1 eV [34,35]. We used Equations (1) and (2) to calculate the values of E_LUMO_ and E_HOMO_:E_HOMO_ (eV) = −|e|(E^ox^_onset, Fc/Fc+_ + 5.1)(1)
E_LUMO_ (eV) = −|e|(E^red^_onset, Fc/Fc+_ + 5.1)(2)

The obtained E_LUMO_ values of both compounds were above the energy level of TiO_2_ semiconductor (−4.2 eV) [38], while the E_HOMO_ values were below the I^−^/I_3_^−^ (−4.9 eV) level [39]. Thus, the electrochemical characteristics obtained for the **KEA** dyes allowed us to state that these compounds are promising for the creation of solar cells.

### 2.6. Photovoltaic Performance

The new dyes based on benzo[*d*][1,2,3]thiadiazole were used to construct DSSCs with a two-layer photoanode made of TiO_2_ (4.5 μm scattering layer and 9 μm transparent layer), I^−^/I_3_^−^ electrolyte, and a Pt film as the counter electrode (See Appendix A for the preparation of DSSCs). The resulting cells were tested under AM1.5 G irradiation (100 mW cm^−2^); the photovoltaic parameters of the best devices compared to the **MAX114** and **WS-2** isomeric dyes are shown in Table 7, and the current-voltage characteristics of the **KEA** series compounds are shown in Figure 6 (statistics of photovoltaic performance of DSSCs fabricated with **KEA** dyes are given in Appendix A). Despite higher values of the molar extinction coefficient in the UV-Vis spectrum obtained for the **WS-2** dye, its *J_SC_* is somewhat lower than for the benzo[*d*][1,2,3]thiadiazole analogue of **KEA321**. At the same time, for a pair of compounds based on the hexahydro-1*H*-carbazole donor block, the trend in *J_SC_* changes is somewhat more predictable: the dye with benzo[*c*][1,2,5]thiadiazole acceptor **MAX114**, which has a four times higher value of ε, showed *J_SC_* of 0.5 mA cm^−2^, which is higher than its **KEA337** isomer. The *V_OC_* values for DSSCs based on the new **KEA** series are higher than those for isomeric analogs with a symmetric acceptor (**WS-2** and **MAX114**), probably, due to reduced recombination through the blocking effect of the dye.

## 3. Materials and Methods

### 3.1. Materials and Reagents

All chemicals were purchased from the commercial sources (Sigma-Aldrich, MO, USA) and used as received. 4,7-Dibromobenzo[*d*][1,2,3]thiadiazole **1** [27], 6-(4,4,5,5-tetramethyl-1,3,2-dioxaborolan-2-yl)-9-(*p*-tolyl)-2,3,4,4a,9,9a-hexahydro-1*H*-1,4-methanocarbazole **3a** [40], 6-(4,4,5,5-tetramethyl-1,3,2-dioxaborolan-2-yl)-9-(*p*-tolyl)-2,3,4,4a,9,9a-hexahydro-1*H*-carbazole **3c** [37], 7-(4,4,5,5-tetramethyl-1,3,2-dioxaborolan-2-yl)-4-(*p*-tolyl)-1,2,3,3a,4,8b-hexahydrocyclopenta[*b*]indole **3d** [41], (9-hexyl-4b,8a-dihydro-9*H*-carbazol-3-yl)boronic acid **2b** [42], phenylboronic acid **2e** [43], thiophen-2-ylboronic acid [44] were prepared according to the previously described protocols. All synthetic operations were performed under a dry argon atmosphere. The solvents were purified by distillation over the appropriate drying agents.

### 3.2. Analytical Instruments

The solution UV-visible absorption spectra were recorded using an OKB Spektr SF-2000 UV/Vis/NIR spectrophotometer (Saint-Petersburg, Russia) controlled with SF-2000 software. All samples were measured in a 1 cm quartz cell at room temperature with a 5.5 × 10^−5^ mol/mL concentration in DCM. The melting points were determined on a Kofler hot-stage apparatus and were uncorrected. ^1^H and ^13^C NMR spectra were taken with a Bruker AM-300 machine (Bruker Ltd., Moscow, Russia) with TMS as the standard. *J* values are given in Hz. MS spectra (EI, 70 eV) were obtained with a Finnigan MAT INCOS 50 instrument (Thermo Finnigan LLC, San Jose, CA, USA). High-resolution MS spectra were measured on a Bruker micrOTOF II instrument using electrospray ionization (ESI). The measurement was operated in a positive ion mode (interface capillary voltage −4500 V) or in a negative ion mode (3200 V); the mass range was from *m*/*z* 50 to *m*/*z* 3000 Da; external or internal calibration was performed with Electrospray Calibrant Solution (Fluka Chemicals Ltd., Gillingham, UK). A syringe injection was used for solutions in acetonitrile, methanol, or water (flow rate 3 μL·min^−1^). Nitrogen was applied as a dry gas; the interface temperature was set at 180 °C. IR spectra were measured with a Bruker “Alpha-T” instrument (Bruker, Billerica, MA, USA) in KBr pellets. Electrochemical measurements were carried out in a dry argon atmosphere using an IPC Pro MF potentiostat (Econix, Russia). The redox properties of compounds were determined using cyclic voltammetry in a three-electrode electrochemical system. A three-electrode system consisting of platinum as the working electrode with an area of 0.8 mm^2^, platinum wire as the counter electrode, and a saturated calomel electrode (SCE) as the reference electrode was employed. The reduction and oxidation potentials were determined in MeCN or in DMF, using 0.1 mol L^−1^ *n*-Bu_4_NClO_4_ as the supporting electrolyte. The cyclic voltammetry (CV) measurements were performed with the use of scan rates of 0.1–5.0 V s^−1^. The first reduction/oxidation potentials were referenced to the internal standard redox couple Fc/Fc^+^. Ferrocene was added to each sample solution at the end of the experiment and employed for calibration.

### 3.3. X-ray Analysis

X-ray diffraction data were collected at 100 K on a four-circle Rigaku Synergy S diffractometer equipped with a HyPix600HE area-detector (kappa geometry, shutterless ω-scan technique), using graphite monochromatized Cu K_α_-radiation. The intensity data were integrated and corrected for absorption and decay by the CrysAlisPro program [45]. The structure was solved by direct methods using SHELXT and refined on *F*^2^ using SHELXL-2018 [46] in the OLEX2 program [47]. All non-hydrogen atoms were refined with individual anisotropic displacement parameters. All hydrogen atoms were placed in ideal calculated positions and refined as riding atoms with relative isotropic displacement parameters. A rotating group model was applied for methyl groups. The Cambridge Crystallographic Data Centre contains the supplementary crystallographic data for this paper No. CCDC 2112091 (**7e**), 2116443 (**7f**), 2061783 (**7b**), 2061784 (**8b**). These data can be obtained free of charge via http://www.ccdc.cam.ac.uk/conts/retrieving.html (accessed on 3 February 2022) (or from the CCDC, 12 Union Road, Cambridge CB2 1EZ, UK; Fax: +44-1223-336033; E-mail: deposit@ccdc.cam.ac.uk). Crystal data and structure refinement for these compounds are given in Table 8 and Table 9.

### 3.4. General Procedure for Suzuki-Miyamura Coupling Reactions

In 50 mL round-bottom flask, of boronic acid **2** or pinacolate ester **3** (0.51 mmol), 4,7-dibromobenzo[*d*][1,2,3]thiadiazole **1** (149 mg, 0.51 mmol) were dissolved in tetrahydrofuran (20 mL), and a solution of K_2_CO_3_ (70 mg, 0.51 mmol) in water (5 mL) was added. The mixture was degassed for 20 min with a stream of argon, and Pd(PPh_3_)_4_ (29 mg, 0.026 mmol) was added. After refluxing for 8 h, the reaction mixture was diluted by ethylacetate (20 mL) and washed with water (3 × 20 mL). Organic layer was dried by Na_2_SO_4_, and then evaporated. The precipitate was purified by column chromatography with mixture of petroleum ether and dichloromethane (15:1).


*7-Bromo-4-(9-hexyl-9H-carbazol-3-yl)benzo[d][1,2,3]thiadiazole*
*(*
**7a**
*)*


Yellow-green oil (59 mg, 25%). ^1^H NMR (300 MHz, CDCl_3_, δ, ppm): 8.57 (s, 1H), 8.08 (d, *J* = 7.7, 1H), 8.00 (d, *J* = 8.5, 1H), 7.76 (d, *J* = 7.8, 1H), 7.60 (d, *J* = 7.9, 1H), 7.49–7.32 (m, 3H), 7.22–7.13 (m, 1H), 4.25 (t, *J* = 7.2, 2H), 1.90–1.74 (m, 2H), 1.41–1.13 (m, 6H), 0.86–0.70 (m, 3H). ^13^C NMR (75 MHz, CDCl_3_, δ, ppm): 155.7, 145.5, 141.1, 140.9, 138.2, 132.1, 128.1, 127.7, 127.5, 126.2, 123.5, 123.1, 122.0, 120.8, 119.4, 109.2, 109.1, 109.0, 43.4, 31.7, 29.1, 27.1, 22.7, 14.1. HRMS-ESI (*m/z*): calcd for (C_24_H_22_BrN_3_S) [M + H]^+^ 464.0790, found 464.0791. UV-Vis (CH_2_Cl_2_, λ_max_, nm/logε): 284/4.24, 376/3.75. IR, ν, cm^−1^: 3057, 2925, 2855, 1729, 1599, 1457, 1347, 1266, 1151, 801, 740. R_f_ = 0.63 (petroleum ether/ethylacetate–5/1). 


*4-Bromo-7-(9-hexyl-9H-carbazol-3-yl)benzo[d][1,2,3]thiadiazole*
*(*
**8a**
*)*


Yellow-green oil (38 mg, 16%). ^1^H NMR (300 MHz, CDCl_3_, δ, ppm): 8.24 (s, 1H), 8.06 (d, *J* = 7.7, 1H), 7.83 (d, *J* = 7.8, 1H), 7.67–7.56 (m, 2H), 7.50–7.36 (m, 3H), 7.23–7.15 (m, 1H), 4.26 (t, *J* = 7.2, 2H), 1.92–1.75 (m, 2H), 1.37–1.19 (m, 6H), 0.85–0.78 (m, 3H). ^13^C NMR (75 MHz, CDCl_3_, δ, ppm): 157.2, 142.9, 141.2, 140.8, 134.9, 131.5, 130.1, 128.6, 126.6, 124.8, 123.8, 122.7, 120.7, 119.6, 119.3, 115.5, 109.6, 109.3, 43.5, 31.7, 29.1, 27.1, 22.7, 14.1. HRMS-ESI (*m/z*): calcd for (C_24_H_22_BrN_3_S) [M + H]^+^ 464.0790, found 464.0791. UV-Vis (CH_2_Cl_2_, λ_max_, nm/logε): 288/4.55, 346/4.00. IR, ν, cm^−1^: 3055, 2925, 2858, 1727, 1599, 1460, 1343, 1253, 1152, 807, 741. R_f_ = 0.52 (petroleum ether/ethylacetate–5/1).


*4,7-Bis(9-hexyl-9H-carbazol-3-yl)benzo[d][1,2,3]thiadiazole*
*(*
**9a**
*)*


Yellow-green oil (26 mg, 8%). ^1^H NMR (300 MHz, CDCl_3_, δ, ppm): 8.76 (d, *J* = 1.4, 1H), 8.44 (d, *J* = 1.6, 1H), 8.19 (m, 3H), 7.96 (s, 2H), 7.83 (d, *J* = 6.7, 1H), 7.61–7.44 (m, 6H), 7.33–7.26 (m, 2H), 4.37 (t, *J* = 7.2, 4H), 1.99–1.88 (m, 4H), 1.49–1.30 (m, 12H), 0.93–0.87 (m, 6H). ^13^C NMR (75 MHz, CDCl_3_, δ, ppm): 157.4, 142.4, 141.2, 141.2, 140.8, 140.6, 136.9, 133.2, 131.3, 128.6, 128.3, 128.0, 127.9, 126.4, 126.0, 125.0, 123.7, 123.5, 123.3, 122.9, 122.0, 120.8, 120.7, 119.5, 119.4, 119.2, 109.5, 109.2, 109.0, 109.0, 43.5, 31.7, 29.2, 27.2, 22.7, 14.1. HRMS-ESI (*m/z*): calcd for (C_42_H_42_N_4_S) [M] 634.3122, found 634.3125. UV-Vis (CH_2_Cl_2_, λ_max_, nm/logε): 296/4.56, 386/4.11. IR, ν, cm^−1^: 3355, 2925, 2857, 1726, 1601, 1456, 1381, 1235, 1147, 799, 724. R_f_ = 0.33 (petroleum ether/ethylacetate–10/1). 


*7-Bromo-4-(9-(p-tolyl)-2,3,4,4a,9,9a-hexahydro-1H-1,4-methanocarbazol-6-yl)benzo[d][1,2,3]thiadiazole*
*(*
**7b**
*)*


Yellow orange solid (57 mg, 23 %), mp 104–105 °C. ^1^H NMR (300 MHz, CDCl_3_, δ, ppm): 7.80–7.75 (m, 2H), 7.66 (d, *J* = 8.3, 1H), 7.55 (d, *J* = 8.0, 1H), 7.26–7.24 (m, 2H), 7.18–7.16 (m, 2H), 6.93 (d, *J* = 8.3, 1H), 4.34 (d, *J* = 8.2, 1H), 3.41 (d, *J* = 8.2, 1H), 2.46 (s, 1H), 2.44 (s, 1H), 2.35 (s, 3H), 1.65–1.55 (m, 3H), 1.47–1.37 (m, 1H), 1.31–1.19 (m, 2H). ^13^C NMR (75 MHz, CDCl_3_): 155.3, 150.6, 145.4, 140.3, 137.6, 134.1, 132.3, 132.0, 130.0, 129.7, 126.8, 126.4, 120.9, 108.2, 107.3, 71.8, 50.3, 43.8, 41.0, 32.5, 28.7, 25.3, 21.0. HRMS-ESI (*m/z*): calcd for [M] 488.0750 found 488.0746. UV-Vis (CH_2_Cl_2_, λ_max_, nm/logε): 310/4.27, 425/4.16. IR, ν, cm^−1^: 3589, 3569, 3448, 1604, 1513, 1460, 1255, 1113, 804. R_f_ = 0.54 (petroleum ether/ethylacetate–10/1). 


*4-Bromo-7-(9-(p-tolyl)-2,3,4,4a,9,9a-hexahydro-1H-1,4-methanocarbazol-6-yl)benzo[d][1,2,3]thiadiazole*
*(*
**8b**
*)*


Yellow orange solid (60 mg, 24 %), mp 207–208 °C. ^1^H NMR (300 MHz, CDCl_3_, δ, ppm): 7.83 (d, *J* = 7.8, 1H), 7.53 (d, *J* = 7.8, 1H), 7.33 (s, 1H), 7.29–7.14 (m, 5H), 6.87 (d, *J* = 8.3, 1H), 4.34 (d, *J* = 8.2, 1H), 3.36 (d, *J* = 8.2, 1H), 2.45 (s, 1H), 2.41 (s, 1H), 2.35 (s, 3H), 1.66–1.52 (m, 3H), 1.47–1.38 (m, 1H), 1.29–1.14 (m, 2H). ^13^C NMR (75 MHz, CDCl_3_) 157.2, 150.7, 141.8, 140.0, 134.8, 134.4, 132.7, 131.4, 130.1, 127.3, 126.9, 123.7, 121.1, 114.6, 107.3, 71.8, 50.2, 43.9, 41.0, 32.5, 28.7, 25.3, 21.0. HRMS-ESI (*m/z*): calcd for [M] 488.0750 found 488.0746. UV-Vis (CH_2_Cl_2_, λ_max_, nm/logε): 314/428, 345/4.27, 410/3.99. IR, ν, cm^−1^: 3856, 3823, 2956, 2926, 2869, 1727, 1603, 1514, 1458, 1271, 1122, 806. R_f_ = 0.41 (petroleum ether/ethylacetate–10/1). 


*4,7-Bis(9-(p-tolyl)-2,3,4,4a,9,9a-hexahydro-1H-1,4-methanocarbazol-6-yl)benzo[d][1,2,3]thiadiazole*
*(*
**9b**
*)*


Orange solid (40 mg, 23 %), mp 140–141 °C. ^1^H NMR (300 MHz, CDCl_3_, δ, ppm): 7.83 (m, 1H), 7.71 (m, 3H), 7.41 (s, 1H), 7.34 (d, *J* = 8.3, 1H), 7.30–7.15 (m, 8H), 6.98 (d, *J* = 8.3, 1H), 6.92 (d, *J* = 8.3, 1H), 4.34 (d, *J* = 8.1, 2H), 3.48–3.34 (m, 2H), 2.48–2.43 (m, 4H), 2.35 (s, 6H), 1.67–1.52 (m, 7H), 1.47–1.39 (m, 2H), 1.32–1.19 (m, 3H). ^13^C NMR (75 MHz, CDCl_3_): 171.3, 157.1, 150.1, 141.3, 140.6, 140.3, 135.7, 134.6, 134.0, 132.3, 132.0, 132.0, 131.9, 130.0, 129.9, 129.5, 129.5, 127.2, 126.8, 126.7, 126.6, 126.5, 123.8, 120.8, 120.6, 107.5, 71.7, 60.5, 50.3, 43.9, 41.0, 32.6, 31.1, 28.7, 25.4, 25.3, 21.2, 21.0, 14.4. HRMS-ESI (*m/z*): calcd for [M] 683.3158 found 683.3158. UV-Vis (CH_2_Cl_2_, λ_max_, nm/logε): 306/4.32, 355/4.51, 433/4.35. IR, ν, cm^−1^: 3569, 3547, 3448, 2750, 2443, 1605, 1513, 1468, 1255, 805. R_f_ = 0.40 (petroleum ether/ethylacetate–10/1).


*7-Bromo-4-(9-(p-tolyl)-2,3,4,4a,9,9a-hexahydro-1H-carbazole-6-yl)benzo[d][1,2,3]thiadiazole*
*(*
**7c**
*)*


Yellow solid (61 mg, 25 %), mp 115–116 °C. ^1^H NMR (300 MHz, CDCl_3_, δ, ppm): 7.82 (s, 1H), 7.78 (d, *J* = 7.9, 1H), 7.70 (d, *J* = 8.3, 1H), 7.57 (d, *J* = 7.9, 1H), 7.28–7.21 (m, 4H), 6.89 (d, *J* = 8.3, 1H), 4.19 (m, 1H), 3.36 (m, 1H), 2.41 (s, 3H), 1.89–1.38 (m, 8H). ^13^C NMR (75 MHz, CDCl_3_): 140.3, 137.9, 137.3, 135.5, 133.8, 132.6, 132.0, 131.8, 130.2, 130.1, 129.4, 128.2, 127.1, 126.9, 124.9, 123.4, 108.9, 108.4, 64.9, 40.6, 28.3, 25.9, 22.9, 21.3, 21.1. HRMS-ESI (*m/z*): calcd for (C_25_H_22_BrN_3_S) [M + H]^+^ 476.0796 found 476.0791. UV-Vis (CH_2_Cl_2_, λ_max_, nm/logε): 303/4.18, 414/4.05. IR, ν, cm^−1^: 3459, 3437, 2928, 2853, 2353, 1606, 1512, 1454, 1272, 929, 805. R_f_ = 0.69 (petroleum ether/ethylacetate = 10/1). 


*4-Bromo-7-(9-(p-tolyl)-2,3,4,4a,9,9a-hexahydro-1H-carbazole-6-yl)benzo[d][1,2,3]thiadiazole*
*(*
**8c**
*)*


Yellow oil (66 mg, 27 %). ^1^H NMR (300 MHz, CDCl_3_, δ, ppm): 7.76 (d, *J* = 7.8, 1H), 7.46 (d, *J* = 7.9, 1H), 7.32–7.26 (m, 1H), 7.23 (m, 2H), 7.21–7.12 (m, 4H), 4.12–4.01 (m, 1H), 3.22 (m, 1H), 2.30 (s, 3H), 1.75–1.38 (m, 8H). MS-MALDI (*m/z*): [M]^+^ calcd for (C_25_H_22_BrN_3_S) 475.07123, found 475.07061.


*4,7-Bis(9-(p-tolyl)-2,3,4,4a,9,9a-hexahydro-1H-carbazole-6-yl)benzo[d][1,2,3]thiadiazole*
*(*
**9c**
*)*


Orange solid (40 mg, 24 %), mp 125–127 °C. ^1^H NMR (300 MHz, CDCl_3_, δ, ppm): 7.88 (s, 1H), 7.81–7.72 (m, 3H), 7.50 (s, 1H), 7.41 (d, *J* = 9.7, 1H), 7.31–7.21 (m, 8H), 6.93 (d, *J* = 8.2, 1H), 6.87 (d, *J* = 8.2, 1H), 4.26–4.15 (m, 2H), 3.45–3.29 (m, 2H), 2.42 (s, 6H), 1.98–1.57 (m, 13H), 1.51–1.42 (m, 3H). ^13^C NMR (75 MHz, CDCl_3_): 157.1, 149.9, 149.8, 141.4, 140.6, 140.3, 136.1, 136.0, 135.3, 133.7, 133.4, 132.3, 130.6, 130.2, 130.1, 130.0, 129.2, 128.0, 127.3, 127.2, 126.9, 126.5, 124.9, 123.3, 123.1, 122.2, 110.6, 110.0, 109.0, 108.9, 64.9, 40.6, 40.6, 28.3, 28.3, 26.0, 25.9, 22.9, 22.8, 21.3, 21.3, 21.2, 21.0, 21.0. HRMS-ESI (*m/z*): calcd for (C_44_H_42_N_4_S) [M + H]^+^ 659.3208 found 659.3203. UV-Vis (CH_2_Cl_2_, λ_max_, nm/logε): 345/4.42, 420/4.24. IR, ν, cm^−1^: 3456, 2926, 2852, 1605, 1513, 1464, 1375, 1268, 1108, 807. R_f_ = 0.38 (petroleum ether/ethylacetate = 10/1). 


*7-Bromo-4-(4-(p-tolyl)-1,2,3,3a,4,8b-hexahydrocyclopenta[b]indol-7-yl)benzo[d][1,2,3]thiadiazole*
*(*
**7d**
*)*


Orange oil (49 mg, 21 %). ^1^H NMR (300 MHz, CDCl_3_, δ, ppm): 7.79 (s, 1H), 7.78 (d, *J* = 6.01, 1H), 7.68 (d, *J* = 8.4, 1H), 7.55 (d, *J* = 7.9, 1H), 7.26–7.16 (m, 4H), 7.00 (d, *J* = 8.4, 1H), 4.93–4.81 (m, 1H), 4.00–3.88 (m, 1H), 2.36 (s, 3H), 2.07–1.60 (m, 6H). ^13^C NMR (75 MHz, CDCl_3_, δ, ppm): 155.4, 149.2, 145.4, 140.1, 137.7, 135.6, 132.1, 132.1, 130.0, 129.6, 126.9, 126.4, 126.3, 120.8, 108.3, 107.5, 69.6, 45.5, 35.4, 33.8, 29.8, 24.6. HRMS-ESI (*m/z*): [M + H]^+^calcd for (C_24_H_20_BrN_3_S) 462.0634, found 462.0634. UV-Vis (CH_2_Cl_2_, λ_max_, nm/logε): 309/3.99, 428/3.84. IR, ν, cm^−1^: 3561, 3434, 2924, 2854, 1605, 1513, 1459, 1378, 1266, 926, 801. R_f_ = 0.53 (petroleum ether/ethyl acetate–10:1).


*4-Bromo-7-(4-(p-tolyl)-1,2,3,3a,4,8b-hexahydrocyclopenta[b]indol-7-yl)benzo[d][1,2,3]thiadiazole*
*(*
**8d**
*)*


Orange oil (51 mg, 22 %). ^1^H NMR (300 MHz, CDCl_3_, δ, ppm): 7.82 (s, 1H), 7.73 (d, *J* = 7.9, 1H), 7.65 (m, 1H), 7.54 (d, *J* = 7.9, 1H), 7.30–7.10 (m, 4H), 7.02 (d, *J* = 3.5, 1H), 5.29 (m, 1H), 4.58 (q, *J* = 6.6, 1H), 2.45 (s, 3H), 2.07–1.59 (m, 6H). MS-MALDI (*m/z*): [M]^+^ calcd for (C_24_H_20_BrN_3_S) 463.07123, found 463.02299.


*4,7-Bis(4-(p-tolyl)-1,2,3,3a,4,8b-hexahydrocyclopenta[b]indol-7-yl)benzo[d][1,2,3]thiadiazole*
*(*
**9d**
*)*


Dark-orange oil (42 mg, 13 %). ^1^H NMR (300 MHz, CDCl_3_, δ, ppm): 7.95 (m, 1H), 7.85 (m, 1H), 7.80–7.67 (m, 4H), 7.22–7.09 (m, 8H), 6.99 (d, *J* = 8.4, 1H), 6.92 (d, *J* = 8.2, 1H), 5.33 (m, 2H), 4.61 (m, 2H), 2.47 (s, 6H), 2.03–1.84 (m, 12H). MS-MALDI (*m/z*): [M]^+^ calcd for (C_42_H_38_N_4_S) 630.28117, found 630.27999.


*7-Bromo-4-phenylbenzo[d][1,2,3]thiadiazole*
*(*
**7e**
*)*


Yellow-white solid (70 mg, 47 %), mp 134–135 °C. ^1^H NMR (300 MHz, CDCl_3_, δ, ppm): 7.98–7.90 (m, 2H), 7.87 (d, *J* = 7.8, 1H), 7.63–7.46 (m, 4H). ^13^C NMR (75 MHz, CDCl_3_): 155.4, 145.5, 137.1, 136.8, 132.0, 129.9, 129.1, 128.9, 128.2, 110.5. HRMS-ESI (*m/z*): calcd for (C_12_H_7_BrN_2_S) [M + H]^+^ 290.9586 found 290.9586. UV-Vis (CH_2_Cl_2_, λ_max_, nm/logε): 279/3.97, 343/3.84. IR, ν, cm^−1^: 3518, 3462, 3438, 3054, 1630, 1455, 1358, 1296, 1117, 931, 845, 771, 693. R_f_ = 0.31 (tetrachloromethane). 


*4,7-Diphenylbenzo[d][1,2,3]thiadiazole*
*(*
**9e**
*)*


Yellow solid (19 mg, 26 %), mp 137–138 °C. ^1^H NMR (300 MHz, CDCl_3_, δ, ppm): 8.04 (d, *J* = 7.6, 1H), 7.94 (d, *J* = 7.6, 1H), 7.86 (m, 1H), 7.74 (d, *J* = 7.5, 1H), 7.68–7.46 (m, 8H). ^13^C NMR (75 MHz, CDCl_3_): 157.2, 142.4, 140.2, 137.5, 133.7, 131.5, 130.0, 129.6, 129.5, 129.3, 129.0, 128.8, 128.8, 128.3, 127.9, 127.5, 127.3, 116.6. HRMS-ESI (*m/z*): calcd for (C_18_H_12_N_2_S) [M + H]^+^ 289.0798 found 289.0794. UV-Vis (CH_2_Cl_2_, λ_max_, nm/logε): 279/3.84, 343/3.72. IR, ν, cm^−1^: 3055, 3026, 2916, 2853, 1542, 1461, 1443, 1352, 1258, 934, 849, 761, 696. R_f_ = 0.29 (tetrachloromethane). 


*7-Bromo-4-(thiophen-2-yl)benzo[d][1,2,3]thiadiazole*
*(*
**7f**
*)*


Yellow-white solid (66 mg, 44 %), mp 133–134 °C. ^1^H NMR (300 MHz, CDCl_3_, δ, ppm): 8.18 (d, *J* = 3.7, 1H), 7.81–7.71 (m, 2H), 7.53 (d, *J* = 5.1, 1H), 7.23 (t, *J* = 4.8, 1H). ^13^C NMR (75 MHz, CDCl_3_): 153.5, 145.7, 138.3, 132.0, 129.9, 129.2, 128.5, 128.3, 125.9, 109.5. HRMS-ESI (*m/z*): calcd for (C_10_H_5_BrN_2_S_2_) [M + H]^+^ 296.9147 found 296.9150. UV-Vis (CH_2_Cl_2_, λ_max_, nm/logε): 306/3.56, 360/3.46. IR, ν, cm^−1^: 3436, 2364, 1625, 1447, 1343, 1267, 1227, 1187, 1017, 931, 826, 688. R_f_ = 0.56 (petroleum ether/ethylacetate–10/1). 


*4,7-Di(thiophen-2-yl)benzo[d][1,2,3]thiadiazole*
*(*
**9f**
*)*


Orange solid (35 mg, 23%), mp 118–119 °C. [29] 

### 3.5. Synthesis of Dyes **KEA321** and **KEA337**

Compounds **7c** or **7d** (0.25 mmol) and **5** (167 mg, 0.38 mmol) were dissolved in toluene (20 mL). The mixture was degassed for 20 min with a stream of argon under stirring, and PdCl_2_(PPh_3_)_2_ (15 mg, 0.013 mmol) was added. After refluxing for 8 h, the reaction mixture was treated with 2 N HCl (20 mL) for 1 h, washed with water (5 × 25 mL), and the organic phase was dried over MgSO_4_, filtered, and concentrated under reduced pressure. The residue was purified by flash chromatography with ethyl acetate/petroleum ether 1:5. The product was used in the next step without further purification. The crude product was dissolved in toluene (20 mL), the cyanoacetic acid tert-butyl ester (53 mg, 0.37 mmol), ammonium acetate (30 mg, 0.39 mmol) and acetic acid (1 mL) were added, and the mixture was refluxed with molecular sieve 4 Å for 6 h. After refluxing, the reaction mixture was diluted with water (25 mL) and extracted with ethyl acetate (4 × 25 mL). The combined organic phases were washed with a sodium carbonate solution (3 × 50 mL), dried over MgSO_4_, filtered, and evaporated under reduced pressure. The residue is purified by silica column chromatography with ethyl acetate/petroleum ether 1:100. *Tret*-butyl ethers was dissolved in solution of CF_3_CO_2_H/CHCl_3_ (5 mL, 1:3). After stirring for 5 h under argon at room temperature, the mixture was diluted with dichloromethane (25 mL) and washed with water (3 × 25 mL). The combined organic phases were dried over MgSO_4_, filtered, and evaporated under reduced pressure. The residue was purified by silica column chromatography with methanol/dichloromethane 1:5 to obtain final products **KEA**.


*2-Cyano-3-(5-(4-(4-(p-tolyl)-1,2,3,3a,4,8b-hexahydrocyclopenta[b]indol-7-yl)benzo[d][1,2,3]thiadiazol-7-yl)thiophen-2-yl)acrylic acid*
*(*
**KEA321**
*)*


Dark-purple powder (133 mg, 95 %), mp 223–225 °C. ^1^H NMR (300 MHz, DMSO-d_6_, δ, ppm): 8.60 (s, 1H), 8.33 (d, *J* = 7.9, 1H), 8.17 (d, *J* = 4.1, 1H), 8.10–7.88 (m, 4H), 7.83 (d, *J* = 8.6, 1H), 7.32–7.21 (m, 3H), 7.00 (d, *J* = 8.5, 1H), 4.98 (m, 1H), 3.95 (m, 1H), 2.32 (s, 3H), 2.17–2.04 (m, 1H), 1.95–1.62 (m, 4H), 1.52–1.40 (m, 1H). ^13^C NMR (75 MHz, DMSO-d^6^, δ, ppm): 163.4, 156.2, 149.6, 148.4, 146.1, 140.9, 140.8, 139.3, 139.1, 137.9, 135.6, 135.2, 131.3, 129.9, 129.8, 129.7, 129.3, 128.7, 128.5, 126.5, 126.2, 125.8, 122.1, 120.0, 116.4, 106.8, 68.5, 44.6, 35.0, 33.1, 24.0, 20.5. MS-MALDI (*m/z*): [M]^+^ calcd for (C_32_H_24_N_4_O_2_S_2_) 560.13439, found 560.13352. UV-Vis (CH_2_Cl_2_, *λ*_max_, nm/log*ε*): 399/4.47, 501/4.36. IR, ν, cm^−1^: 3434, 2979, 2931, 2217, 1710, 1581, 1429, 1369, 1281, 1257, 1149, 1090, 799. R*_f_* = 0.46 (dichloromethane/methanol–2:1). 


*2-Cyano-3-(5-(4-(9-(p-tolyl)-2,3,4,4a,9,9a-hexahydro-1H-carbazol-6-yl)benzo[d][1,2,3]thiadiazol-7-yl)thiophen-2-yl)acrylic acid*
*(*
**KEA337**
*)*


Dark-purple powder (133 mg, 93 %), mp 234–236 °C. ^1^H NMR (300 MHz, DMSO-d_6_, δ, ppm): 8.27 (m, 2H), 8.02–7.72 (m, 5H), 7.45–7.37 (m, 1H), 7.27 (m, 3H), 6.83 (d, *J* = 8.1 Hz, 1H), 4.33–4.15 (m, 1H), 2.89–2.75 (m, 1H), 2.35 (s, 3H), 1.96–1.84 (m, 2H), 1.78–1.61 (m, 2H), 1.57–1.28 (m, 4H). ^13^C NMR (75 MHz, DMSO-d_6_, δ, ppm): 163.1, 156.2, 149.4, 146.2, 141.1, 139.5, 138.9, 137.5, 137.2, 136.7, 134.9, 132.8, 130.1, 129.9, 129.5, 128.0, 126.5, 126.2, 124.7, 123.3, 122.8, 122.4, 118.5, 110.8, 109.6, 108.0, 63.7, 27.4, 25.2, 22.5, 22.0, 20.6, 20.5. MS-MALDI (*m/z*): [M]^+^ calcd for (C_33_H_26_N_4_O_2_S_2_) 574.15063, found 574.14917. UV-Vis (CH_2_Cl_2_, *λ*_max_, nm/log*ε*): 454/4.21. IR, ν, cm^−1^: 3435, 2925, 2852, 2213, 1685, 1609, 1514, 1377, 1209, 1139, 802. R*_f_* = 0.46 (dichloromethane/methanol–2:1). 

## 4. Conclusions

In summary, two new organic D-A-π-A1 dyes of the **KEA** series were designed and synthesized. In order to prepare the precursors for these dyes, i.e., unsymmetrical monoaryl(hetaryl) derivatives of benzo[*d*][1,2,3]thiadiazole, the Suzuki-Miyamura cross-coupling reaction between 4,7-dibromobenzo[*d*][1,2,3]thiadiazole and aryl(hetaryl)boronic acids and their esters was thoroughly investigated. Fused indoline and carbazole derivatives formed two isomers, whereas phenyl- and 2-thienylboronic acids formed only one 4-substituted compound. Cyclic voltammetry measurements showed that fused indoline and carbazole had much higher E_HOMO_ values than benzene and thiophene, which indicated that they were much stronger donors, which should facilitate electron transfer in the intermediate complex, contributing to the formation of a mixture of monoisomers, in contrast to thienyl- and phenylboronic acids. The structure of mono derivatives was rigorously proven by NMR spectroscopy and X-ray analysis. The **KEA** dyes had a very narrow energy gap and the position of E_HOMO_ and E_LUMO_ was suitable for the effective use of these dyes in solar cells. The asymmetric nature of benzo[*d*][1,2,3]thiadiazole **KEA** dyes in comparison with the symmetrical benzo[*c*][1,2,5]thiadiazole isomers led to a hypsochromic shift in the ICT band and lower extinction coefficients which suggested that that the symmetry breaking of the acceptor benzo[*d*][1,2,3]thiadiazole fragment reduced the efficiency of intramolecular charge transfer for compounds of the **KEA** series. **KEA** dyes had a narrow HOMO-LUMO gap of 1.5–1.6 eV, which apparently led to the better power efficiency (PCE), i.e., 5.17%, for ****KEA321**,** which was superior to the corresponding symmetrical benzo[*c*][1,2,3]thiadiazole dyes **WS-2** and **MAX114** (5.07 and 4.90%). Thus, it was shown that the exchange of a symmetrical benzo[*c*][1,2,5]thiadiazole ring to an asymmetrical benzo[*d*][1,2,3]thiadiazole can lead to promising photovoltaic properties.

## Data Availability

Not applicable.

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
