# Peer review of "Novel D-A-π-A1 Type Organic Sensitizers from 4,7-Dibromobenzo[d][1,2,3]thiadiazole and Indoline Donors for Dye-Sensitized Solar Cells"

_molecules, 2022, doi:10.3390/molecules27134197_

Round 1
Reviewer 1 Report
The article: "Novel D-A-π-A1 type organic sensitizers from 4,7-dibromo- 2 benzo[d][1,2,3]thiadiazole and indoline donors for dye-sensitized solar cells" presents the synthesis of two new organic dyes, type D-A-π-A1 of the KEA series together with their characteristics and description of individual synthesis steps with identification of intermediate products. Finally, dyes were used to prepare DSSC solar cells, and the results of photovoltaic measurements were compared to literature data obtained for WS-2 and MAX114 dyes.
In my opinion, the article may be interesting to Readers of "Molecules", but it needs to be corrected. Some parts of the work are described too poorly and require additional explanations or additional experiments.
11. Abstract: There are too many dye symbols that make it difficult to read the article, especially at the beginning, when the reader does not yet know the content of the entire article.
22. Figure 4: How will the Authors explain the non-zero absorbance (and thus the molar absorption coefficient) for KEA337 in the range of ~600 - ~800 nm? Has the phenomenon of aggregation of dyes in solution been studied by Authors?
33. Why was dichloromethane used to study the absorption properties of dyes? Was dichloromethane also used to sensitize dyes by the preparation of DSSC solar cells?
44. At Table 5, the Authors state that the absorption maxima were determined at a dye concentration of 5.5*10-5 M. Meanwhile, in the content of the work (line 239, page 9) a value of 1*10-5 M was given for KEA321 and KEA327 dyes. What are the reasons for these discrepancies?
55. Why was DMF used as a solvent by the electrochemical studies?
66. How were the DSSC cells prepared? The description of the experimental part is missing. What dye solution (concentration) was used to sensitization? What was the exact composition and concentration of the electrolyte components?
77. Table 7 shows the best values obtained for DSSC cells. Error analysis is missing. How many cells of a given series have been prepared to find the best values?
88. Figure 6.: Current density-voltage curves of DSSC – this is probably not such an important drawing to show in the main part of the work. I suggest to move to the supplement.
99. Was the comparison of dyes (literature data) WS-2 and MAX114 in DSSC systems (Table 7) measured under the same conditions (e.g. electrolyte composition, counter-electrode, dye concentration, sensitization time, type of titanium dioxide paste, thickness of mesoporous layer)? Optimally, the Authors should prepare in the same way the solar cells with dyes WS-2 and MAX114, measured under the same conditions to be able to draw conclusions for DSSC cells with new dyes.
110. Page 11, line 308-309. "reduced recombination through the blocking effect of the dye" Have the authors conducted any research in this direction? For example, using electrochemical impedance spectroscopy?
111. Part 4. Materials and Methods, point 3.2 (should be 4.2). It is: "All samples were measured in a 1cm quartz cell at room temperature with a 3,7*10-5M concentration in DMSO". Meanwhile, the Authors in the paper included absorption spectra of dyes in dichloromethane. At the same point: "the luminescence spectra were recorded...." - I did not find in the article the results of luminescent measurements. Please make sure that the further parts of the description under "Analytical Instruments" are correct.
112. I suggest to create a Supplement and transfer some of the results (eg. Synthesis data and characteristics of intermediate products). In my opinion, the article is too long.
Author Response
Response to Reviewer 1.
The authors are grateful to the reviewer for a highly professional review.
Reviewer 1:
Abstract: There are too many dye symbols that make it difficult to read the article, especially at the beginning, when the reader does not yet know the content of the entire article.
Authors:
Bringing general formulas for dyes could increase the volume of the abstract very much without increasing the understanding of the meaning of the paper, so we would prefer to leave the dye abbreviations in the text of the abstract.
Reviewer 1:
Figure 4: How will the Authors explain the non-zero absorbance (and thus the molar absorption coefficient) for KEA337 in the range of ~600 - ~800 nm? Has the phenomenon of aggregation of dyes in solution been studied by Authors?
Authors:
The phenomenon of aggregation was not investigated by us, it is quite likely that it is observed in this case.
Reviewer 1:
Why was dichloromethane used to study the absorption properties of dyes? Was dichloromethane also used to sensitize dyes by the preparation of DSSC solar cells?
Authors:
Cells were constructed using 0.3 mM concentration in toluene/EtOH (v/v, 7/3) dye solutions.
Reviewer 1:
At Table 5, the Authors state that the absorption maxima were determined at a dye concentration of 5.5*10-5 M. Meanwhile, in the content of the work (line 239, page 9) a value of 1*10-5 M was given for KEA321 and KEA327 dyes. What are the reasons for these discrepancies?
Authors:
Solutions with a concentration of 5.5*10-5 M were used; the error has been corrected in the text of the paper.
Reviewer 1:
Why was DMF used as a solvent by the electrochemical studies?
Authors:
Final dyes dissolve best in DMF.
Reviewer 1:
How were the DSSC cells prepared? The description of the experimental part is missing. What dye solution (concentration) was used to sensitization? What was the exact composition and concentration of the electrolyte components?
Authors:
The general manufacturing procedure and characteristics of the DSSCs have been added to the Supporting Information.
Reviewer 1:
Table 7 shows the best values obtained for DSSC cells. Error analysis is missing. How many cells of a given series have been prepared to find the best values?
Authors:
Statistics of photovoltaic performance of DSSCs fabricated with KEA dyes have been added to the Supporting Information
Reviewer 1:
Figure 6.: Current density-voltage curves of DSSC – this is probably not such an important drawing to show in the main part of the work. I suggest to move to the supplement.
Authors:
We believe that current density-voltage curves of DSSCs is important information that is present in almost all papers devoted to organic solar cells, so its presence in the text of the paper seems necessary to us.
Reviewer 1:
Was the comparison of dyes (literature data) WS-2 and MAX114 in DSSC systems (Table 7) measured under the same conditions (e.g. electrolyte composition, counter-electrode, dye concentration, sensitization time, type of titanium dioxide paste, thickness of mesoporous layer)? Optimally, the Authors should prepare in the same way the solar cells with dyes WS-2 and MAX114, measured under the same conditions to be able to draw conclusions for DSSC cells with new dyes.
Authors:
The measurement conditions and method for creating DSSCs based on WS-2 and MAX114 were similar to those we used for the KEA series dyes. DSSCs based on WS-2 and MAX114 were constructed by us earlier, before starting work on the submitted manuscript. The photovoltaic characteristics for cells based on WS-2 and MAX114 presented in the manuscript were taken from previously published papers.
Reviewer 1:
Page 11, line 308-309. "reduced recombination through the blocking effect of the dye" Have the authors conducted any research in this direction? For example, using electrochemical impedance spectroscopy?
Authors:
Unfortunately, we are unable to provide experimental data characterizing the recombination kinetics inside the devices. Nevertheless, based on a wide array of literature data, including our earlier published works, we can make an assumption that the asymmetric structures of the KEA series, in comparison with dyes with a symmetrical benzo[c][1,2,5]thiadiazole acceptor WS-2 and MAX114 provide cells based on them with more pronounced the resistance for charge recombination and, as a result, resulting in higher electron density in the TiO2 conduction band with a higher Voc output (see, for example, Wu, Y.; Zhang, X.; Li, W.; Wang, Z.-S.; Tian, H.; Zhu, W. Hexylthiophene-Featured D-A-π-A Structural Indoline Chromophores for Coadsorbent-Free and Panchromatic Dye-Sensitized Solar Cells. Adv. Energy Mater. 2012, 2, 149–156, doi:10.1002/aenm.201100341; Wu, Y.; Zhu, W. Organic sensitizers from D–π–A to D–A–π–A: effect of the internal electron-withdrawing units on molecular absorption, energy levels and photovoltaic performances. Chem. Soc. Rev. 2013, 42, 2039–2058, doi:10.1039/C2CS35346F; Tanaka, E.; Mikhailov, M.S.; Gudim, N.S.; Knyazeva, E.A.; Mikhalchenko, L. V.; Robertson, N.; Rakitin, O.A. Structural features of indoline donors in D–A-π-A type organic sensitizers for dye-sensitized solar cells. Mol. Syst. Des. Eng. 2021, 6, 730–738, doi:10.1039/D1ME00054C).
Reviewer 1:
Part 4. Materials and Methods, point 3.2 (should be 4.2). It is: "All samples were measured in a 1cm quartz cell at room temperature with a 3,7*10-5M concentration in DMSO". Meanwhile, the Authors in the paper included absorption spectra of dyes in dichloromethane. At the same point: "the luminescence spectra were recorded...." - I did not find in the article the results of luminescent measurements. Please make sure that the further parts of the description under "Analytical Instruments" are correct.
Authors:
Corrected as suggested by reviewer.
Reviewer 1:
I suggest to create a Supplement and transfer some of the results (eg. Synthesis data and characteristics of intermediate products). In my opinion, the article is too long.
Authors:
Synthesis data and characteristics of intermediate products, in our opinion, constitute a big and important part of this paper, therefore we prefer not to shorten the synthetic part of the paper. The Supplementary Materials was submitted with the paper.
Reviewer 2 Report
The herein manuscript presents a thorough investigation of asymmetrical derivatives of [1,2,3]thiadiazole bearing indoline and carbazole moieties on one end and thiophene-cyanoacrylate on the other, as fully organic-metal free dyes for DSSCs. The synthetic methodology and the protocols and characterization of the molecules is exhaustive and solid. The optical and electrochemical characterization of the materials provides sufficient proof of the dyes applicability as sensitizers and the actual evaluation of DSSC based on these dyes clearley depicts the chemical structure effect on the dyes efficiency and device performance.
Only minor spelling errors are found in the text that should be corrected prior to publication.
Overall the herein manuscript is well written and scientifically solid, thus deserving publication in Molecules.
Author Response
Response to Reviewer 2.
The authors are grateful to the reviewer for a kind and highly professional review.
Reviewer 2:
Only minor spelling errors are found in the text that should be corrected prior to publication.
Authors:
We have carefully checked the manuscript again and corrected spelling errors.
Round 2
Reviewer 1 Report
The Authors answered all my questions from the review. I suggest to publish the article in Molecules.